# RECAP: REwriting Conversations for Intent Understanding in Agentic Planning

## Abstract

Understanding user intent is essential for effective planning in conversational assistants, particularly those powered by large language models (LLMs) coordinating multiple agents. However, real-world dialogues are often ambiguous, underspecified, or dynamic, making intent understanding a persistent challenge. Traditional classification-based approaches struggle to generalize in open-ended settings, leading to brittle interpretations and poor downstream planning. We propose RECAP (REwriting Conversations for Agent Planning) [1], a new benchmark designed to evaluate and advance intent rewriting, reframing user-agent dialogues into concise representations of user goals. RECAP captures diverse challenges such as ambiguity, intent drift, vagueness, and mixed-goal conversations. Alongside the dataset, we introduce an LLM-based evaluator that compares planning utility given a user-agent dialogue. Using RECAP, we develop a prompt-based rewriting approach that outperforms baselines, in terms of plan preference. We further demonstrate that fine-tuning two DPO-based rewriters yields additional utility gains. Our results highlight intent rewriting as a critical and tractable component for improving agentic planning in open-domain dialogue systems.

## 1 Introduction

Understanding user intent is a foundational challenge in building effective conversational assistants, particularly in systems that rely on the coordinated use of multiple agents and tools to complete complex tasks (Xu et al., 2024b; Song et al., 2023a; Wang et al., 2024a). Agentic planning (Wang et al., 2023; Li et al., 2025; Erdogan et al., 2025) allows these systems to autonomously decompose and sequence tasks, enabling agents to determine the most effective actions and coordination strategies to achieve user goals. In such systems, accurate intent detection is essential for successful planning, as the system must decide what action to take and how best to delegate or execute it across agents. Misinterpreting user intent can result in planning errors, a degraded user experience, and inefficient task completion.

In real-world multi-turn conversations, user intent is rarely static or perfectly stated (Zhou et al., 2024). Users may revise goals mid-conversation, introduce ambiguous or incomplete commands, or digress into side topics. These natural phenomena of user-agent dialogue, such as vagueness, intent drift, and ellipsis, pose significant challenges for current planning modules that rely on a clear and up-to-date understanding of user goals. Traditional approaches to intent understanding, such as intent classification, often rely on a fixed schema of predefined intents and slots (Goo et al., 2018; Budzianowski et al., 2018). While effective in narrow domains, these approaches struggle with open-ended or evolving conversations common in LLM-powered assistants (Arora et al., 2024). Such methods are susceptible to intent drift within conversation, fail to generalize to unseen or out-of-domain queries, and often force user inputs into rigid categories that do not reflect their actual goals. These limitations make it difficult for downstream planning modules to act on user input with the necessary flexibility and accuracy. More adaptive strategies are, hence, needed to handle the fluid, underspecified, and dynamic nature of real human intent in open-domain systems.

One promising strategy is intent rewriting: introducing a module that rephrases the user-agent dialogue into a concise, clarified representation of the user's most recent intent (Galimzhanova et al.,

---

[1]We provide code & data in supplementary materials.

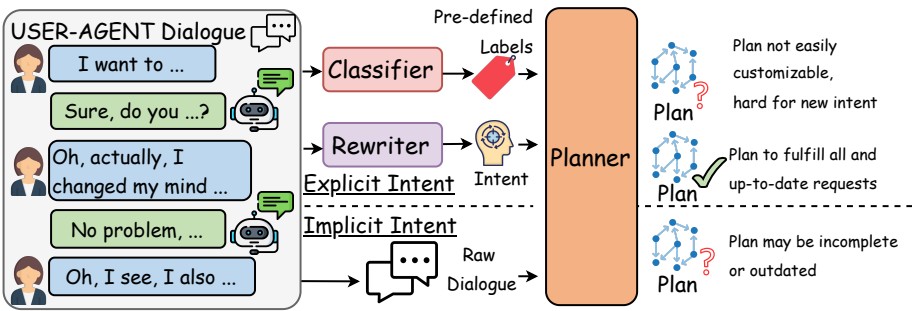

Figure 1: When implemented effectively, explicit intent modeling tends to produce higher-quality plans, particularly in fulfilling user requests and capturing multiple or evolving intents. In contrast, planning based on implicit intent is more prone to incomplete or outdated understanding, especially in longer conversations. Among methods for modeling intent, rewriting provides a more flexible approach than intent classification, enabling support for underrepresented and compound intents.

2023). This rewritten intent distills the relevant context, removes distractions, resolves ambiguity, and refocuses the system on the core user goal. By providing a cleaner target for action, intent rewrites enable downstream planners to make better decisions with less reliance on the full dialogue history.

Despite the growing interest in task-oriented dialogue and agent planning (Byrne et al., 2019; King & Flanigan, 2024; Xu et al., 2024a; Qiao et al., 2025; Gan et al., 2025), there remains a lack of benchmarks specifically designed to evaluate intent rewriting in this context. Existing datasets either focus narrowly on slot-filling and task completion (Budzianowski et al., 2018) or treat rewriting as a standalone summarization problem (Li et al., 2023), without grounding it in agent behavior or planning effectiveness. As a result, there is limited empirical understanding of what makes a rewrite effective for agent planning.

To bridge this gap, we introduce RECAP (REwriting Conversations for Agent Planning), a new benchmark that systematically captures diverse intent rewriting challenges across domains, including under-specified, drifted intent and multi-intent conversations. Alongside this dataset, we provide an effective LLM-based evaluator that judges the quality of agent plans given dialogue history and rewrites. Using RECAP, we develop a prompt-based intent rewriter that consistently outperforms baseline approaches, in terms of downstream plan preference. Building on this, we fine-tune two DPO (Rafailov et al., 2023)-based rewriters starting from our best-performing zero-shot model, achieving equivalent or better utility in **77.8%** of the cases[2].

## 2 EXPLICIT INTENT MODELING FOR PLANNING

Many task-oriented applications, such as virtual assistants, engage users through dialogue interfaces and increasingly rely on multi-agent collaboration behind the scenes to decompose and execute complex tasks. This architecture demands accurate and adaptable intent understanding, as well as effective agent planning. As illustrated in Figure 1, we assume the presence of a base chat agent that conducts multi-turn conversations with the user, maintaining a trajectory of USER–AGENT dialogue. Notably, the chat agent does not directly solve the task itself, but instead keeps the conversation flowing by presenting intermediate results generated by the underlying multi-agent system.

Complementing the chat agent is a planner that interprets user intent from the dialogue history up to the current point and generates a plan to coordinate action agents in order to complete the task (e.g., searching the web, drafting an email, creating a file). The planner produces a structured plan, represented as a Directed Acyclic Graph (DAG), which captures the sequence and dependencies of sub-tasks required to achieve the user's goal. Each node in the DAG represents a sub-task, while

---

[2]combining win & tie rate of our DPO-based rewriter, as shown in Table 5.

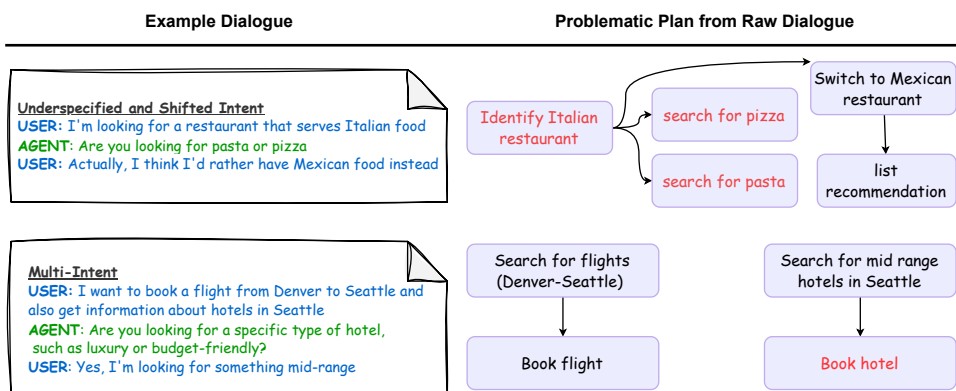

Figure 2: Qualitative examples of short dialogues with complex intents that confuse the planners when provided in raw form. Red nodes highlight issues in the generated plans.

edges define the logical flow between them. In this paper, the planner is implemented using state-of-the-art LLMs and carefully written prompts (more details in Section 5.1).

While it may seem straightforward to feed the entire conversation history directly to the planner and rely on it to infer the implicit user intent, this approach can be problematic, particularly in real-world settings where User–Agent interactions are often noisy and include irrelevant or ambiguous turns. Specifically, we identify four common challenges in everyday User–Agent conversations that can lead to confusion or failure in planning: ***underspecified intent***, where the user's goal lacks sufficient detail; ***noisy input***, where irrelevant or off-topic dialogue turns obscure the main objective; ***shifted intent***, where the user changes their goal mid-conversation; and ***multi-intent***, where multiple distinct goals are presented simultaneously or sequentially without clear separation.

Figure 2 presents qualitative examples of short dialogues with complex intents that confuse planners when processed in raw form. In the first dialogue, the user initially mentions an interest in Italian restaurants but later shifts to searching for a Mexican restaurant. The plan generated from the raw dialogue incorrectly interprets the chat agent's suggestions (e.g., pizza and pasta) as user requests and fails to recognize that the user's original intent is no longer relevant. In the second example, the user wants to book a flight but only seeks information about hotels. The planner, given the full dialogue without explicit intent modeling, mistakenly proceeds to book both the flight and a hotel. With explicit intent modeling, the correct interpretation would be: *"search for a Mexican restaurant"* and *"book a flight from Denver to Seattle and gather information about mid-range hotels in Seattle."* While these examples are brief due to space constraints, such confusion is far more frequent in longer, more complex dialogues.

Quantitatively, we observe notable differences in preference, semantics, and structure between plans generated from raw conversation history and those generated from rewritten inputs. These discrepancies are consistent across multiple planning models, including the reasoning-capable `o3-mini`, highlighting the importance of clear and well-structured intent representations for effective agent planning. We present detailed results in Section 5.2.

## 3 RECAP BENCHMARK

Existing agent planning benchmarks either assume clearly defined tasks with well-specified requirements (e.g., TravelPlanner (Xie et al., 2024)) or focus solely on vague or underspecified intent (e.g., IN3 (Qian et al., 2024)). As discussed in Section 2, additional challenges such as intent shifts and nuanced details can lead to suboptimal downstream planning. To enable a deeper understanding of how to effectively represent complex user intent, we introduce RECAP, a benchmark designed to evaluate the ability of conversational rewriters to capture accurate, unambiguous, up-to-date, and comprehensive intent for downstream multi-agent planning.

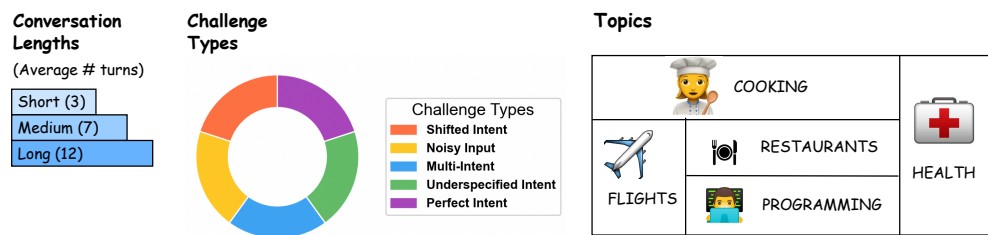

Figure 3: RECAP Dataset Characteristics

## 3.1 DATASET CONSTRUCTION

Our goal is to construct a diverse and challenging dataset of user-agent conversations for intent understanding in planning tasks. We synthetically generate two-way dialogues that reflect realistic user-agent interactions. Specifically, we create conversations that span a variety of topics (*cooking, programming, health, flights, restaurants*), conversation lengths (*short, medium, and long*), and intent understanding categories (*shifted intent, noisy input, underspecified intent, multi-intent, perfect intent*), as illustrated in Figure 3. This design allows our dataset to capture a wide range of scenarios relevant to planning tasks based on understanding complex user intent.

Human-generated datasets are often costly to produce, and recent work has demonstrated the promise of LLM-generated synthetic data (Kim et al., 2025), especially for intent understanding tasks (Maheshwari et al., 2024). Hence, we adopt a prompt-based generation approach (see Section A.2) using LLMs (`GPT-4o` OpenAI (2024) and `LLaMA 3.3-70B` Meta (2024)) to simulate a back-and-forth conversation on a given topic between a user and a chat agent. Conversations are also designed to be challenging in at least one of the predefined categories.

The generated dialogues undergo careful human vetting to ensure they are coherent, adhere to the assigned topic and challenge type, and follow the specified conversation-length constraints. We also filter out any dialogues in which the chat agent hallucinates or attempts to solve the user's task. Additionally, since intent analysis is performed only on user utterances, we require each conversation to end with a user turn, and discard any that violate this constraint. In total, RECAP comprises 810 validated conversation instances (see Section F).

## 3.2 EVALUATION METRICS

Having constructed a set of challenging user-agent conversations, we apply various rewriters to each conversation and feed the resulting rewritten intent into a planner to generate the final task plans. We evaluate the quality of these plans in a pair-wise method using three main categories of metrics.

**Structural Metrics**  To capture structural differences between the plan DAGs, we compute the following metrics:
*Node and Edge Count Differences*: $\Delta_{\text{nodes}} = N_1 - N_2$, $\Delta_{\text{edges}} = E_1 - E_2$, where $N_i$ and $E_i$ denote the number of nodes and edges in plan $P_i$, respectively.
*Graph Edit Distance* (Sanfeliu & Fu, 1983): $\text{GED}(P_1, P_2)$, which measures the minimum cost of edit path to transform plan $P_1$ to $P_2$ such that they are isomorphic.
These metrics provide a quantitative view of how structurally similar or divergent two plans are.

**Semantic Metrics**  We assess the semantic distance between generated plans using `BERTScore` (Zhang et al., 2020).
Specifically, we compute `Semantic Distance` as: $1 - \text{BERTScore}(P_1, P_2)$, where $P_1$ and $P_2$ are the two plans being compared.

**Preference Metric**  As the ultimate measure of utility, we assess whether the planner produces the most effective plan given a rewritten intent. In this work, we employ human annotators as well as utilize LLM-based evaluators, who are asked to judge plan preference on the following rubrics:

- **Latest Intent:** The plan should reflect the user's most recent goals or intent as expressed in the conversation.

- **Fabrication:** The plan should avoid unnecessary, repetitive, or fabricated steps.

- **Task Granularity:** The plan should offer specific and detailed actions.

- **Task Completeness:** The plan should include all necessary steps to accomplish the goal.

- **Logical Order:** Tasks should be arranged in a coherent, logical sequence. Parallelizable tasks should be grouped accordingly for efficiency.

We employ a pairwise comparison setup: two rewritten intents from the same source conversation are each fed into the planner, producing two separate plans. Human annotators, following the rubric above, are shown both plans (in randomized order) and asked to select the one that better aligns with the user's intended goal. If both are judged equally effective (or ineffective), a tie is recorded.

More on the implementation details of all metrics and human evaluation study is described in Section C.2.

## 3.3 LLM-AS-JUDGE EVALUATOR

While human evaluation provides high-quality preference signals, it is both costly and time-consuming. To mitigate this, we explore the feasibility of training models to predict human preferences between pairs of plans. As a baseline, we prompt a frozen large language model (LLM) to select the preferred plan in a zero-shot setting, mirroring the structure of the human annotation task.

Beyond this, we fine-tune two preference models using the collected human labels on `RECAP-train` with the majority vote of the human preference labels obtained through the evaluation process described later in Section 5.3. These models take as input a source conversation along with two candidate plans and are trained to predict the preferred plan or indicate a tie. Further implementation and sampling details are mentioned in Section C.2.2.

## 4 RECAP REWRITERS

### 4.1 CONSTRUCTING REWRITES

We begin by introducing two baseline rewriters used to evaluate the impact of rewriting quality on downstream planning.

`Dummy` rewriter simply reproduces the original multi-turn USER-AGENT conversation verbatim, without any modification or abstraction. This baseline allows us to observe how the planner responds to raw, unprocessed dialogue input.

LLM-based `Basic` rewriter performs a direct summarization of the full conversation history using a generic summarization prompt (Section B). This approach does not receive any specific instructions regarding which parts of the conversation are important to preserve, such as intent shifts or irrelevant contents. As a result, the summary may omit critical information required for accurate planning, making it a useful reference point for assessing the added value of more targeted rewriting approaches.

To capture the nuanced aspects of query rewriting, we adopt a prompt-based generation approach (see Section B) using `GPT-4o` (OpenAI, 2024) with a temperature setting of 0. This setup is used to generate high-quality rewrites optimized for downstream planning, which we refer to as the `Advanced` rewriter.

The `Advanced` rewriter produces a refined and task-aware representation of the original multi-turn conversation. Unlike generic summarization, it is explicitly prompted to produce rewrites that are concise, unambiguous, and well-aligned with the user's most recent goals. It emphasizes fine-grained aspects of intent understanding, such as detecting the latest user intent(s), filtering out irrelevant or noisy input, and making reasonable assumptions in cases where the user's intent is under-specified. This guided approach allows the rewrite to serve as a more effective interface between the user's dialogue and the planner. Qualitative examples of the different rewrites is shown in Figure 4.

## 4.2 Training Rewriter

To further enhance the performance of the rewriter, we fine-tune the advanced summarizer using *Direct Preference Optimization (DPO)* (Rafailov et al., 2023) and name it `DPO:human` This method leverages human preference annotations on pairs of plans generated from the same source conversation. For each annotated plan pair, we trace back to the corresponding rewrites that produced them. The rewrite that led to the preferred plan is treated as a positive sample, while the other is treated as a negative sample. These preference pairs serve as training signals to fine-tune a `GPT-4o` model, encouraging it to generate rewrites that are more likely to result in plans preferred by humans. This setup enables indirect supervision of the rewriter, without requiring manually curated gold rewrites, by aligning the learning objective with the downstream metric of planning utility.

Because high-quality human preference labels are expensive and limited in quantity, we also train an additional version of the rewriter using pseudo-labels generated by our strongest automated plan preference evaluator. This model (`DPO:LLM`) follows the same DPO training paradigm, offering a scalable but weaker alternative to human-supervised fine-tuning. Training implementation details are further described in Section D.

## 5 Evaluation

### 5.1 Experimental setup

**Planner** To evaluate the impact of different input rewrites on downstream task planning, we adopt a controlled setup using a static LLM-based planner. In this setup, the planner agent does not interact with the user or the environment; instead, it receives a rewritten user intent as input and generates a task plan in the form of a directed acyclic graph (DAG). The raw output from the language model is parsed into a structured graph format (details provided in Section C.1, which allows us to verify the acyclicity of the plan and supports structured analysis. We use `GPT-4o` with `temperature` set to 0, to ensure deterministic generation, minimizing randomness across different runs. The detailed prompting setup used to guide the planner is described in Section C.1.

**Data** For our experiments in the following sections, we uniformly sample and utilize 150 conversation instances from RECAP due to cost constraints (eg. human annotations). We include studies on the entire dataset in Section F. We partition these 150 RECAP conversations into `train`, `val`, and `test` splits with a ratio of 60-10-30. By holding the planner fixed and systematically varying only the rewritten input, we isolate the effect of intent formulation on the resulting task decomposition. This setup enables a controlled evaluation of how different rewriting strategies influence the structure and quality of generated plans.

### 5.2 Sensitivity

We begin with a sensitivity analysis, examining the variability of plans generated using different intent representations, with the aim of assessing planners' sensitivity to their input—namely, user intent specified in various forms. This motivates the need for effective conversation rewriters. For each conversation, we generate two rewrites using `Dummy` (the most naive model) and `Advanced` (our best-performing prompt-based rewriter), simulating two extremes of rewriting[3]. The `Dummy` and `Advanced` rewrites are

Table 1: Percentage of `Dummy` and `Advanced` plans preferred based on human evaluation; "Tie" reports cases when no plan is specifically preferred. With `IN3-70`, most pairs result in ties, whereas `RECAP-toy` exhibits a clear distinction between good and bad intents. The longer the conversations, the more sensitive the planner becomes.

| Length | RECAP-toy | | | IN3-70 | | |
|---|---|---|---|---|---|---|
| | **Short** | **Medium** | **Long** | **Short** | **Medium** | **Long** |
| `Dummy` | 26.67 | 20.00 | 16.67 | 16.67 | 8.62 | 33.33 |
| Tie | 23.33 | 20.00 | 20.00 | **66.67** | **70.69** | **66.67** |
| `Advanced` | **50.00** | **60.00** | **63.33** | 16.67 | 20.69 | 0.00 |

---

[3]The `Basic` rewriter is intentionally omitted from this sensitivity analysis, as its quality and impact is expected to lie in between the `Dummy` and `Advanced` rewriters. Full pairwise comparison is in section 5.3

separately provided as input to a static LLM planner (`GPT-4o`, `temperature=0`) using a fixed prompt template (Appendix C.1). Evaluation is conducted on two benchmarks: a 70-instance subset of the `IN3` dataset (Qian et al., 2024), and a synthetic `RECAP-toy` dataset of 70 USER-AGENT dialogues generated with `GPT-4o` (Appendix B), following the same procedure as the main RECAP data generation.

Table 1 shows that human annotators much more frequently prefer plans generated from `Advanced` rewrites on `RECAP-toy`, demonstrating that improved intent formulations lead to better plan quality, even with identical planning models. Beyond human-judged plan quality, we observe consistent trends in more objective structural and semantic metrics. Beyond human-judged plan quality, we see similar trends in objective structural and semantic metrics. Figure 12 in Appendix E.2 shows that these plans diverge structurally—in node/edge counts and graph edit distance—especially as conversation length increases, highlighting greater sensitivity in complex dialogues.

In contrast to RECAP, `IN3` exhibits lower sensitivity across all metrics. Human preferences are more often tied, and structural and semantic differences are reduced regardless of conversation length. This indicates that `IN3` lacks the realism and complexity to surface input-sensitivity effects, reinforcing the need for more challenging datasets like RECAP. To confirm these results are not planner-specific, we replicate the experiments on `RECAP-toy` using `LLaMA 3.3-70B` and `GPT-o3-mini` (Figures 10, 11). All models exhibit consistent sensitivity to rewrites under identical prompt and decoding settings.

## 5.3 COMPARING REWRITERS

Building on findings from Section 5.2, we evaluate the performance of the prompt-based rewriters introduced in Section 4.1 in generating rewrites that support effective plan generation. The analysis is conducted on conversations from `RECAP-train`[4].

**Plan Preference** Results in Table 2 highlight that plans derived from `Advanced` rewriter are consistently preferred across most intent-related challenges. This effect is particularly strong in conversations involving complex or evolving intents, eg. *Shifted Intent* and *Multi-Intent*. In *Shifted Intent* contexts, the `Basic` summarizer is frequently outperformed, as unguided summarization often omits important details of the users' requests. `Dummy` performs better as it is provided with the full context, but the planner remains vulnerable to outdated intents and is less effective than `Advanced`. In *Under-specified Intent* scenarios, however, `Advanced` is slightly outperformed by `Basic` because in the benchmark we deliberately include conversations with so-called "fake intent shifts" (see Appendix B.1 for examples) where the user appears to initiate a new request but is in fact continuing to refine a previous one. In such cases, `Advanced`

Table 2: Win/Tie/Loss percentage for each rewriter grouped by challenge. Each rewriter competes against all other rewriters.

| Challenge | Rewriter | Win Rate | Tie Rate | Loss Rate |
|---|---|---|---|---|
| Shifted Intent | Dummy | 21.43 | 59.52 | 19.05 |
| | Basic | 2.38 | 47.62 | 50.0 |
| | Advanced | **50.0** | 45.24 | 4.76 |
| Noisy Input | Dummy | 23.81 | 54.76 | 21.43 |
| | Basic | 11.90 | 54.76 | 33.33 |
| | Advanced | **30.95** | 57.14 | 11.90 |
| Multi-Intent | Dummy | 14.29 | 47.62 | 38.09 |
| | Basic | 19.05 | 52.38 | 28.57 |
| | Advanced | **40.48** | 52.38 | 7.14 |
| Underspecified Intent | Dummy | 12.5 | 55.0 | 32.5 |
| | Basic | **20.0** | 70.0 | 10.0 |
| | Advanced | 17.50 | 75.0 | 7.50 |
| Perfect Intent | Dummy | 11.36 | 63.64 | 25.0 |
| | Basic | 15.91 | 72.73 | 11.36 |
| | Advanced | **20.46** | 68.18 | 11.36 |
| **Total** | Dummy | 16.67 | 56.19 | 27.14 |
| | Basic | 13.81 | 59.52 | 26.67 |
| | Advanced | **31.90** | 59.52 | 8.57 |

may incorrectly interpret these as true shifts, leading to inaccurate rewrites.

These results underscore the pivotal role of input formulation in determining plan quality, showing a well-guided rewriter can convey the appropriate amount of information to the downstream planner, neither omitting critical details nor overwhelming its reasoning. At the same time, they expose

---

[4]The prompt-based rewriters are zero-shot and not trained for this task; we use the training partition to avoid contaminating the held-out test set used later for evaluating trained rewriters.

the limitations of purely prompt-based approaches, motivating our subsequent experiments with a trained rewriter presented in Section 5.5.

**Structural and Semantic Comparisons:**
As shown in Table 3, plans derived from different rewrites exhibit noticeable structural divergence. Notably, GED is highest between plans generated from `Basic` and `Advanced` rewrites respectively, indicating that these input variants induce markedly different planning behaviors. Despite using identical prompts and models, such structural shifts reflect the planner's high sensitivity to surface form and implicit signals in the input.

Table 3: Average structural and semantic distances between plans generated with prompt-based rewriters.

| Plan Comparison | $\Delta_{nodes}$ | $\Delta_{edges}$ | GED | Sem Dist |
|---|---|---|---|---|
| Dummy vs Basic | 1.68 | 2.18 | 4.99 | 0.10 |
| Dummy vs Advanced | 1.70 | 2.36 | 5.56 | 0.11 |
| Basic vs Advanced | 1.87 | 2.49 | 6.44 | 0.11 |

## 5.4 LEARNING TO PREDICT PLAN PREFERENCE

As discussed in Section 3.3, we explore the use of LLMs to predict human plan preferences, enabling scalable evaluation. We compare baseline and fine-tuned LLM evaluators on train and test splits sampled from `RECAP-train`, enriched with more challenging comparisons between plans from `Advanced` and `DPO:human` rewrites. Full details on setup and sampling methodology are provided in Appendix C.2.2.

Table 4 summarizes performance across of various LLM models. The fine-tuned `gpt-4.1` model achieves the highest accuracy and F1

Table 4: LLM-as-Judge plan preference evaluator, prompted and fine-tuned.

| Model | Train | | | Test | |
|---|---|---|---|---|---|
| | Acc% | F1 | | Acc% | F1 |
| baseline:gpt-4o-mini | 38.91 | 0.31 | | 37.5 | 0.35 |
| baseline:gpt-4o | 36.36 | 0.31 | | 43.75 | 0.39 |
| baseline:gpt-4.1 | 38.55 | 0.38 | | 45.0 | 0.46 |
| ft:gpt-4o-mini | 69.09 | 0.67 | | 48.75 | 0.48 |
| ft:gpt-4o | 72.00 | 0.72 | | 53.75 | 0.48 |
| ft:gpt-4.1 | 74.91 | 0.73 | | 65.01 | 0.65 |

scores on both train and test sets, substantially outperforming zero-shot baselines (`gpt-4o-mini`, `gpt-4o`, and `gpt-4.1`). These results highlight the promise of fine-tuned LLMs as reliable and cost-efficient evaluators in nuanced plan comparison tasks.

## 5.5 EVALUATION OF TRAINED REWRITERS

Next, we compare two DPO-based rewriters (introduced in Section 4.2) against our best-performing `Advanced` rewriter on the held-out `RECAP-test` set, using the static `GPT-4o` planner. The `DPO:human` model is trained using human preference labels from `RECAP-train`, while `DPO:LLM` is trained on the same plan pairs but uses preferences judged by an LLM-as-a-judge evaluator. We employ our best-performing LLM evaluator, a fine-tuned `GPT-4.1`.

As shown in Table 5, `DPO:human` achieves the highest win rate across nearly all intent challenge categories, outperforming the `Advanced`

Table 5: Win/Tie/Loss percentage for `DPO:human` vs `Advanced` and `DPO:LLM` vs `Advanced` rewriters.

| Challenge | Rewriter | Win Rate | Tie Rate | Loss Rate |
|---|---|---|---|---|
| Shifted Intent | DPO:human | **55.56** | 11.11 | 33.33 |
| | DPO:LLM | 22.22 | 33.33 | 44.44 |
| Noisy Input | DPO:human | **44.44** | 33.33 | 22.22 |
| | DPO:LLM | **44.44** | 0.0 | 55.56 |
| Multi-Intent | DPO:human | **44.44** | 33.33 | 22.22 |
| | DPO:LLM | 33.33 | 33.33 | 33.33 |
| Underspecified Intent | DPO:human | **30.0** | 50.0 | 20.0 |
| | DPO:LLM | 20.0 | 60.0 | 20.0 |
| Perfect Intent | DPO:human | **75.0** | 12.50 | 12.50 |
| | DPO:LLM | 25.0 | 62.50 | 12.50 |
| **Total** | DPO:human | **48.88** | 28.90 | 22.22 |
| | DPO:LLM | 28.88 | 33.33 | 37.78 |

rewriter. Notably, it yields substantial gains in more difficult scenarios such as *Shifted Intent*, *Multi-Intent* and *Underspecified Intent*, suggesting that aligning with human preferences helps capture finer nuances of user intent. In contrast, `DPO:LLM` performs competitively in categories

like *Perfect Intent* and *Multi-Intent*, but does not consistently surpass `Advanced` across all intent-understanding categories. This indicates that while LLM-generated supervision offers scalability, it may still fall short of the effectiveness achieved through human preferences. Figures 8 and 9 further illustrate plan preferences across test cases. These results highlight the value of human-aligned supervision for training robust rewriters and demonstrate DPO as a scalable path toward adaptive, human-aligned input reformulation in task-oriented dialogue systems.

## 6 RELATED WORK

**Multi-Turn Intent Understanding**   Intent understanding is a core component of dialogue systems, particularly in multi-turn interactions where user intent can be vague, drift over time, or be obscured by noisy utterances. Traditional intent classification approaches and slot filling solutions in Dialogue State Tracking (DST) works (Budzianowski et al., 2018; Wu et al., 2019; Mrkšić et al., 2017; Rastogi et al., 2020) aim to map user utterances to one or more predefined intent categories, offering clear signals to inform the system's next action. However, these methods rely heavily on a well-defined intent taxonomy and often struggle to generalize across domains. To address these limitations, research on intent discovery and out-of-distribution (OOD) detection has emerged (Song et al., 2023b; Wang et al., 2024b). While these methods aim to identify novel or ambiguous intents, they face challenges such as low precision in distinguishing subtle intent variations and difficulty in adapting to evolving user goals. A more flexible approach is to directly rewrite user intent utterances, without relying on predefined intent classes.

**Query Rewriting**   In information-seeking and Retrieval-augmented Generation (RAG) settings, query rewriting has been shown to enhance retrieval quality by incorporating conversational context. Wu et al. (2022) introduced CONQRR, a transformer-based model trained with reinforcement learning to optimize downstream retrieval rewards. Ye et al. (2023) explored prompting LLMs like GPT-3 to generate context-aware rewrites, showing that LLMs can infer implicit information from earlier turns. Mo et al. (2024) proposed CHIQ, a two-stage method where an LLM first enhances the dialogue history and then rewrites the final user query, achieving strong performance on conversational search tasks. While effective, these approaches are primarily designed for search scenarios and assume a task-agnostic, retrieval-focused environment. Intent rewriting in realistic multi-round conversations for planning and agent coordination remain underexplored.

**LLM-Based Planning**   Recent work has explored LLMs for planning in ambiguous, multi-step dialogue settings. Chen et al. (2025) proposed ACT, a method that trains LLMs to proactively ask clarification questions using a contrastive self-training objective, promoting better discrimination between plausible next steps. Deng et al. (2024) introduced Self-MAP, a memory-augmented planner that uses reflection to adjust plans in response to evolving user goals, showing improved performance on complex instruction-following tasks. Although these approaches show promising signals in reasoning over ambiguity and intent drift, they typically require carefully designed planning solutions involving fine-tuning or the integration of additional components—such as dedicated reflection modules or memory-augmented agents. RECAP provides planner-agnostic benefits by operating independently of the underlying planner's architecture or capabilities and offers a more flexible and interpretable representation.

This gap of flexible intent understanding for agent planning is especially evident in the lack of robust benchmarks that reflect the complexities of real-world conversations. Qian et al. (2024) introduced IN3, a benchmark that captures vague user intents and focuses on generating clarification questions. However, it does not adequately address other challenging scenarios, such as intent shifts or multiple simultaneous intents.

## 7 CONCLUSION

We introduced RECAP, a new benchmark for evaluating intent rewriting in LLM-powered conversational systems, capturing key challenges like ambiguity, drift, and goal shifts. By reframing dialogue into concise intent representations, rewriting enables more accurate and flexible agent planning. Our experiments show that both prompt-based and DPO-trained rewriters significantly improve planning utility, even without explicit preference labels. These results highlight intent rewriting as a promising direction for building more effective and adaptive dialogue agents.

## ETHICS STATEMENT

In this work, we propose a novel benchmark for intent rewriting and understanding for agentic planning. Our dataset was synthetically generated using LLMs which may introduce artifacts or biases inherent to the model used. However, we ensured to vet all generated samples to remove any unwanted instances, and also redact any use of real or fake names and contact information in the generated conversations.

In our evaluation methodology, we made sure that experiments involving human annotators were conducted in accordance with ethical research guidelines. Annotators provided informed consent for participation and the purpose of the task and the manner in which their annotations will be used was clearly communicated.

Artifacts used in our work, including publicly available ones, have been clearly cited and utilized with intended use. We also used commercially available AI models (e.g., GPT) in a manner consistent with their terms of service. These data are intended for research purposes only and do not contain real user information. We mention the portions of this work, such as the dataset construction, rewrite as well as plan generation, and evaluation, which utilize AI models; specifying the implementation details in each case.

Finally, while our findings point toward improved plan quality through rewrite optimization, we caution against over-reliance on such systems without human oversight, particularly in high-stakes or safety-critical domains.

## REPRODUCIBILITY STATEMENT

We provide the data, source code, and further implementation details to reproduce our results including the model used, prompts, and specified hyperparameters.

The data curation and vetting process is clearly described in Section A.2 and Section F. Details about the human annotation process, including the task provided, rubrics, UI interface and methods to compute inter-annotator agreement score, is defined in Section C.2.2.

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

# A   APPENDIX

## A   CONSTRUCTING CONVERSATIONS

In order to suit our study setting, we aim to obtain conversation instances between a USER and an AGENT focused on task-oriented dialogue with intent-related challenges. We utilize the existing IN3 dataset (Qian et al., 2024), which itself is derived from MultiWoz dataset (Budzianowski et al., 2018), as well as synthetically generate our own.

### A.1   CONVERSATION CONSTRUCTION: IN3

Qian et al. (2024) provide an instruction understanding & execution benchmark, where a `task` eg. *"Find a recipe for homemade pizza."* is annotated with a label `vague`, denoting if the task-intent is vague or not. If the task is vague, the benchmark provides `missing details` with an `inquiry` i.e. a clarification question eg. *""Do you have any dietary restrictions or preferences?""* and possible answer `options` to this query eg. *["Gluten-free", "Vegan", "No restrictions"]*

We modify this dataset to build conversations prompting `gpt-4o` with `temperature=0` to convert the initial task and missing details as a USER-AGENT style conversation. The USER begins the conversation with the `task`, and the AGENT follows up with each `inquiry`. The USER answers the inquiry with one of the answer `options` provided, at random. The prompt used is shown in Prompt:A.1.

We perform this method on 70 instances of the IN3 data (to match the instances in RECAP-toy dataset) and filter only those tasks which have been labeled as vague.

---

**Conversation Construction: IN3**

```
You will be provided a task sentence and some missing details as
a list.  Each missing detail has an inquiry and corresponding
options.  Your job will be to convert this to a friendly User-Agent
conversation.  The User begins conversation with the task.  The Agent
responds with each missing detail inquiry one at a time, and the User
responds with the option as response.

Task:  task
Missing Details:  missing_details

Output Format:  Each conversation should a list of strings starting
with 'USER:' or 'AGENT:'.
```

---

### A.2   CONVERSATION CONSTRUCTION: RECAP

To generate a conversation dataset with tougher intent-understanding related challenges, we follow the methodology described in Section 3.1. The prompt used to generate such conversations is detailed in Prompt:A.2 which aims to generate conversations across different topics, conversation lengths and intent-understanding challenges. During simulation, we emphasize that the chat agent should not attempt to solve the user's task.

The topics included are *cooking, programming, health, flights, restaurants*, taking inspiration from existing intent classification works such as Budzianowski et al. (2018).

The conversation length categories are defined as:

**short** : where the total number of USER and AGENT utterances is up to 5

**medium** : where the total number of USER and AGENT utterances more than 5 but up to 10

**long** : where the total number of USER and AGENT utterances more than 10 but up to 20

---

**Conversation Construction: RECAP**

```
Generate a conversation between a USER and an AGENT on the topic:
{topic}.
The USER begins with a task-oriented query.  The AGENT only asks
clarifying or follow-up questions to understand the USER's intent
and constraints.  It must not solve the task.

The conversation should be {conv_len}, stay on-topic, and be
coherent.

Each conversation must end with a USER utterance and no utterance
should include unrelated or off-topic remarks.

The challenge types are:
{challenge_instructions}

Output a single JSON object with challenge names as keys and
conversations as values.
Each conversation is a list of strings starting with 'USER:' or
'AGENT:'.
```

---

Table 6: Example USER-AGENT dialogues with **short** conversation length in the **cooking** domain, illustrating different intent-related challenges.

| Category | Conversation |
|---|---|
| **Shifted Intent** | **USER**: I want to bake a cake for my birthday. 
 **AGENT**: What kind of cake are you thinking of? 
 **USER**: Actually, I'd rather make some fresh chocolate chip cookies. |
| **Noisy Input** | **USER**: Hi, how's it going? I need to cook dinner tonight. 
 **AGENT**: Hello! Sure, I'll be happy to assist you today! I can help you with cooking. What type of dinner are you planning to make? 
 **USER**: Thank you for assisting me! Umm, something with chicken. |
| **Underspecified Intent** | **USER**: I need to cook something for a party. 
 **AGENT**: How many people are you planning to serve? 
 **USER**: Not sure, but I want it to be easy to eat. |
| **Multi-Intent** | **USER**: I want to make a meal that's both healthy and tasty. 
 **AGENT**: Are you looking for a specific cuisine or dietary restriction? 
 **USER**: I'm open to anything, but it should be quick to prepare and not too expensive. |
| **Perfect Intent** | **USER**: I want to make chicken parmesan with spaghetti for 4 people. Do you have a good recipe? 
 **AGENT**: Would you like to use homemade or store-bought marinara sauce? 
 **USER**: I'll use homemade sauce and serve it with a side salad. |

We utilize `gpt-4o` and `llama-v3p3-70b-instruct` (Fireworks) models with `temperature=1` to generate varied and diverse instances. We curate and pick 150 conversations generated using these different models separately, and modify if needed to ensure adherence to prompt instructions. Characteristics of the dataset are illustrated in Figure 3.

Examples of conversations across intent-understanding categories are included in Table 6.

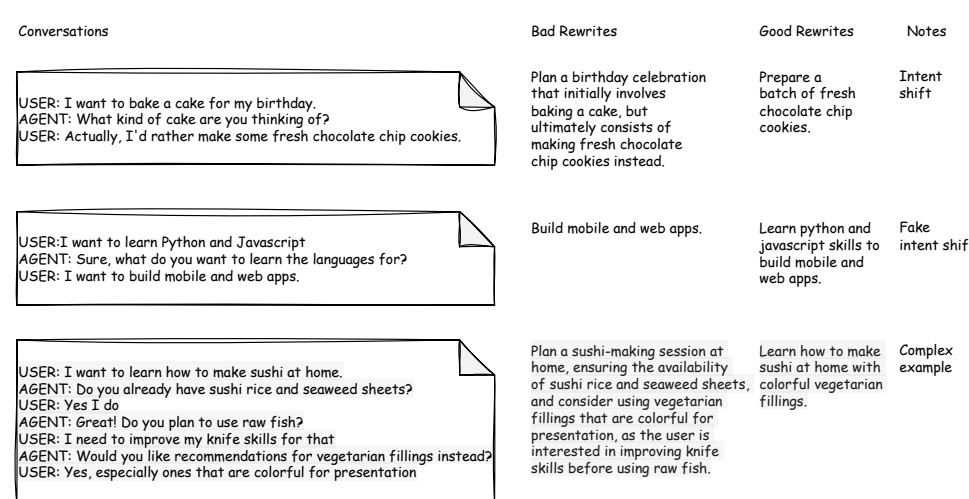

Figure 4: Qualitative examples of good and bad rewrites.

A simplified version of this prompt (using only conversation length as criteria) is used to generate 70 instances for a toy dataset which we use for sensitivity analysis in Section 5.2.

## B    REWRITE GENERATION

Rewrites are generated using `gpt-4o` with `temperature` set to 0. Prompt:B outlines the prompt used to generate rewrites for the `Basic` and `Advanced` summarizers. The dummy rewriter simply outputs the input conversation as a string.

```
Prompt used to Generate Rewrites

Basic Rewriter
Summarize the following USER-AGENT conversation

Conversation:
{conversation}
--------------------------------------------------------------------
Advanced Rewriter
Summarize the following USER-AGENT conversation into a single,
concise sentence describing the user's intended task.
The summary should reflect the final user goal or intent, in an
instruction style.
The user's intent may be changed completely i.e. shifted or it may
be updated with further specifications for the original intent.
Ensure to capture the latest user intent with only the necessary
specifications.
Filter out any noise or irrelevance in the input.
Do not introduce new information. Only include what is stated or
clearly implied, make assumptions only if necessary.

Conversation:
{conversation}
```

### B.1    QUALITATIVE EXAMPLES

Qualitiave examples of the output of different rewriters is shown in Figure 4.

**Fake Intent Shifts**   The first row of Figure 4 shows a simple example where the user intent shifted from baking a cake to making cookies. However, the second row illustrates a more subtle case, which we refer to as a *fake intent shift*. Here, the user appears to start a new intent, but is actually providing further specification of the previous intent to "learn Python and JavaScript". If a rewriter is over-optimized for detecting intent shifts, it may produce an incorrect rewrite such as "building mobile and web apps," which would lead the plan toward app development rather than gathering educational materials.

**Compound Example**   In the third row, rewrites must capture the nuanced aspects of a complex user intent to generate accurate plans. For instance, although the USER mentions "improving knife skills," this is not their main goal, and they later agree to use a vegetarian filling, which reduces the need for knife skills. This nuance is captured by the good rewrite (produced by `Advanced`) and reflected in the corresponding plan, which focuses solely on the user's actual goal. In contrast, the bad rewrite (produced by `Basic` rewriter) is misled by irrelevant details, resulting in a plan that includes "Practice Knife Skills" as a redundant and imprecise task. It also adds a redundant step to "ensure availability of sushi rice and seaweed sheets," which the user had already confirmed in the second turn.

## C   PLAN GENERATION AND EVALUATION

### C.1   GENERATING PLANS

We use the following prompt to generate plans given an input task i.e. output of a rewriter. For RECAP, we use a *static* `gpt-4o` planner with `temperature=0`, so as to obtain as deterministic outputs from the planner as possible.

---
**Prompt used for Generating Plans**

```
You are a planner responsible for creating high-level plans to
solve any task.  Understand the user intent from the input and plan
accordingly.  Consider breaking down complex tasks into subtasks.

Represent your plan as a graph where each node corresponds to a step,
and each edge represents a dependency between two steps.  If a node
requires the output from a previous node as an input, ensure it is
included in the edge list.

The output should be structured in the following JSON format:
'nodes':  <list of JSON nodes with keys 'id':  <node id as integer>,
'name':  <sub-task node name> >,
'edges':  <list of tuples [node_id, node_id]>

Input:
{input}
```
---

After obtaining the plan generated from the LLM, the plan is converted to DAG format using `networkx:MultiDiGraph` utilizing the corresponding nodes and edges.

### C.2   EVALUATING PLANS

In Section 3.2, we defined the three categories of metrics we used to evaluate plans - structural, semantic and preference based.

#### C.2.1   STRUCTURAL & SEMANTIC EVALUATION OF PLANS

**Structural Metrics:**   $\Delta_{\text{nodes}} = N_1 - N_2$ and $\Delta_{\text{edges}} = E_1 - E_2$, are computed using in-built `networkx` functions, which corresponds to the difference in the number of nodes and edges, respectively, between two plans. We use the `optimize_graph_edit_distance` function within `networkx` to compute the graph edit distance between the two plans $\text{GED}(P_1, P_2)$. This measures the minimum cost of edit path (sequence of node and edge edit operations) transforming

**You will be provided with a conversation between a user and a chat agent.**

The user makes an initial query, and the chat agent asks some clarifying questions to better understand the user's intent.

**Note:**

1. In some cases, the user's intent may be fully clear through the conversation (e.g., "For dinner, I want to cook white sauce pasta with chicken").
2. While in other instances, few aspects of the user's intent remain vague (e.g., "I want to have pasta for dinner").
3. In some cases, the user may backtrack on their initial query and ask help regarding a different task (e.g., the user initially asks the agent to help find restaurants with Japanese cuisine, but later decides they want Italian cuisine instead).
4. Furthermore, the user may provide details which may seem as a shift of their intent, but is only a further specification of their previous intent.
5. In some cases, the user may provide multiple tasks to the agent (e.g., "Find me programming resources for Web Development and Mobile App Development").

Now, keeping in mind the **entire conversation** between the user and agent, two **plans** have been generated to **perform actions which shall help fulfil the task** described in the above conversation.

A plan breaks down the user's task(s) into various sub-tasks which are connected with arrows showing a logical flow from one sub-task to another.

**Your task is to choose which plan is better to solve the task.**

Please refer to the rubrics below when conducting the comparison:

- **latest_intent**: A good plan should fulfill the *user's updated goals/intent* from the conversation.
- **fabrication**: A good plan should be accurate and *not include unnecessary, repetitive or false tasks*.
- **task_granularity**: A good plan should provide *more specific and detailed steps*.
- **task_completeness**: A good plan should include *all necessary steps to achieve the user goal*.
- **logical_order**: A good plan should arrange tasks in a *coherent, logical* sequence. If tasks can be done in *parallel*, they should be done so for better efficiency.

Please select your preference: **PlanA**, **PlanB**, or **TIE** if you feel both plans are equally good and capable of solving the task.

Figure 5: Rubrics for Plan Evaluation

plan $P_1$ to $P_2$ such that they are isomorphic. While the generic `graph_edit_distance` function may be computationally expensive and slow, especially for larger graphs, the optimized version helps calculate the nearest approximation of GED for such cases.

**Semantic Metrics:** We combine the text from all task nodes from plan $P_1$ and $P_2$ respectively and report the F1 BertScore (Zhang et al., 2020) between them as
`Semantic Distance` = 1 - BertScore($P_1, P_2$).

### C.2.2 Plan Preference

For each conversation instance, given two plans generated correspondingly from two different rewriters (eg. `Dummy` vs `Basic`), we use human as well as LLM evaluators to measure the pair-wise performance between the two generated plans.

The evaluators are provided a conversation, two plans A and B (when presenting plans A & B to the user, the plans from the rewriters eg. *dummy* and *basic* are randomly shuffled to ensure no positional bias). The evaluators are further provided instructions with criteria to choose the best plan among the two: A, or B, or a tie if both plans are equally good.

It is to be noted that (a) the evaluators are not provided any information about the rewriter (input to planner); and (b) that the plans are generated using a *static* planner (detailed in Section C.1) so as to indirectly measure the impact of the corresponding rewriter on the downstream plan performance/preference.

**Human Annotators:** We recruited 3 expert in-house annotators, who are proficient in English, and currently based in the United States of America, with at least a graduate-level degree. The annotators were clearly explained the objective of the task and how their annotations would be utilized. To measure agreement between the annotators we use average of the pair-wise accuracy scores between each of the annotators. We also note the subjectivity and difficulty of the task, which leads to *moderate* to *good* agreement scores across our human-evaluation studies.

The instructions provided to the human annotators were the same as provided to the LLM Evaluator which is detailed in Figure 5. An example of the interface used for human annotation is shown

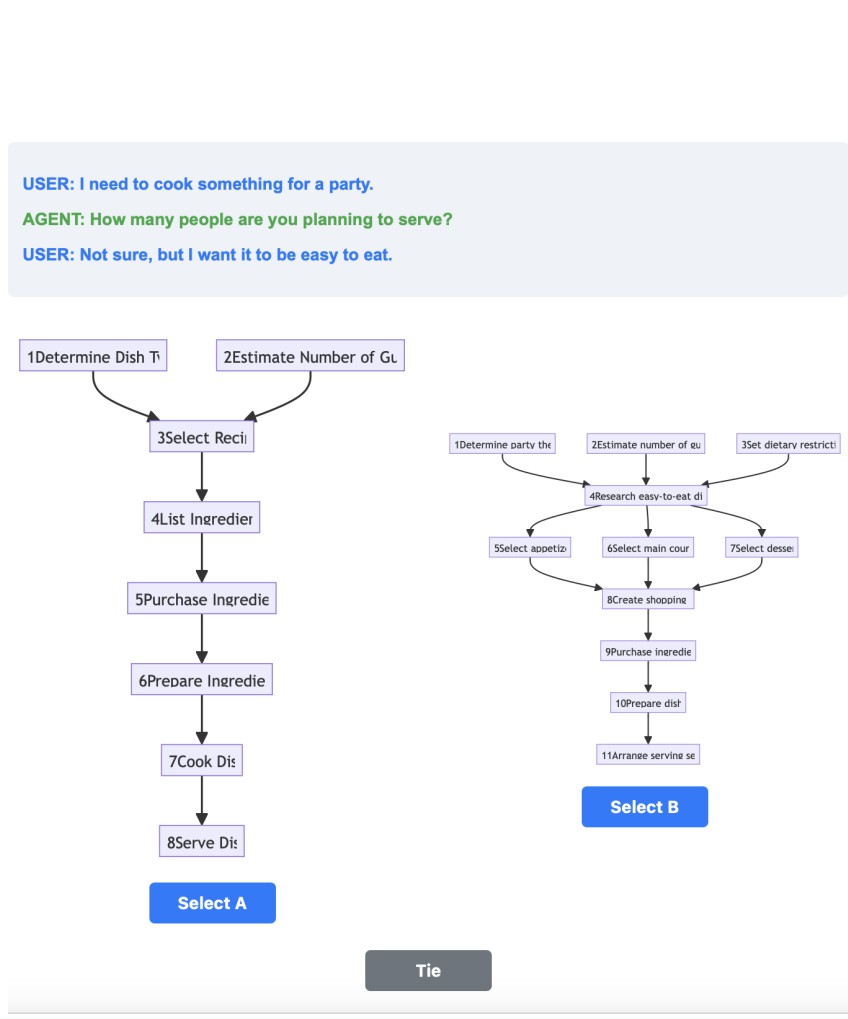

Figure 6: Interface for Human Preference Annotation

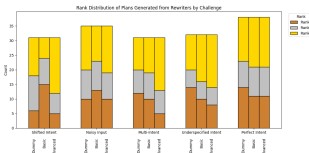 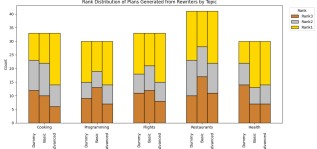 

(a) Human Preference of Plans by Intent Category

(b) Human Preference of Plans by Topic

(c) Human Preference of Plans by Conversation Length

Figure 7: Ranked Analysis: Human Preference of Plans across Rewriters (`Dummy`, `Basic` and `Advanced`) on `RECAP-test`: `Advanced` ranks 1st across all intent categories. Average pairwise inter annotator accuracy: 75.4%.

.

in Figure 6. Once the annotations are obtained, the majority label of the annotators is used as the preference label for the plan-pair.

To compare how plans from different rewriters were preferred by humans, we report the Win/Tie/Loss rates for each rewriter i.e. for all plan-pairs, how many times was the plan from the corresponding rewriter preferred (win), not preferred (loss), or a tie.

We also build a ranking mechanism to rank the 3 plan-pairs per conversation instance. For the three rewrites and corresponding plans i.e. `Dummy`, `Basic` and `Advanced`, a +1 score is given to a rewriter if it is preferred over another, +0.5 given to both rewriters if there is a `TIE`, else 0 is given for losses. The total scores across plan-pairs for a conversation instance are used to rank the performance of these rewriters for that instance, using standard ranking mechanism eg. if `Basic` and `Advanced` both have +2.5 scores while `Dummy` has a score of 0, the ranks are:

`Advanced` rewriter: Rank 1
`Basic` rewriter: Rank 1
`Dummy` rewriter: Rank 3

The results from this ranked analysis is shown in Figures 7a, 7b, 7c, measuring the count that each rewriter was ranked $r_i$ across the different intent-understanding challenges, topics, and conversation lengths in our dataset.

---

**Prompt used for Evaluating Plans**

```
You will be given a task-oriented dialogue between a USER and an
AGENT as well as two plans.  Your task is to choose the plan that
better addresses the user's intent.

Please refer to the rubrics below when conducting the comparison:
{RUBRICS}

The plans are evaluated on their ability to fulfill the above
rubrics.  Both plans are considered equally good when they are
equally capable of fulfilling the above rubrics.  In that case,
output TIE.

Conversation:  {conversation}
Plan A: {planA}
Plan B: {planB}

Which plan better fulfills the user's request?  Reply with 'A', 'B',
or 'TIE'."
```

---

**LLM Evaluator:** Human annotations are not scalable, hence we rely on LLMs as plan-preference evaluators on a large sclae. The LLM evaluator is also prompted with the same instructions as given to the users using Prompt:C.2.2.

To further improve LLM evaluators, we fine tune them on the `RECAP-train` data with the majority vote of the human preference labels obtained earlier. We additionally add 40 samples comparing the `Advanced` vs `DPO:human` plans from Section 5.5 so as to include tougher instances of plan comparison while training our fine-tuned evaluator. These instances are also generated only from conversations included in `RECAP-train`, so as to not contaminate the `RECAP-test` dataset.

These samples (`RECAP-train` + tougher instances) are then split into `train-val-test` splits (60-10-30) for the sole purpose of fine-tuning LLM evaluators. We utilize the same Prompt:C.2.2 as previously to prepare the training, validation and test data. For our baseline, we use a zero-shot approach, prompting models `gpt-4o-mini`, `gpt-4o` and `gpt-4.1`. Furthermore, use OpenAI fine-tuning for each of these models using the human majority label, with hyperparameters: `batch_size`, `learning_rate_multiplier`, and `n_epochs` set to `auto`.

## D    TRAINING REWRITERS USING DPO

---

**Prompt used for Training Rewriters using DPO**

```
You will be given a task-oriented dialogue between a USER and an
AGENT. Your task is to reinterpret or rewrite the conversation in
a format that clearly conveys the USER's intent, optimized for a
downstream planning agent that will decompose the request into
actionable subtasks.
Based on your judgment, you may choose to rewrite the conversation or
retain the original format.

Conversation:  conversation
```

---

In Section 4.2, we described adopting a preference-based learning strategy using Direct Preference Optimization (DPO), where given a pair of plans evaluated, we trace each plan back to its corresponding rewrite. The rewrite responsible for the preferred plan is treated as the `preferred_output`, and the other as the `non_preferred_output`. These preference pairs serve as supervisory signals to fine-tune a `gpt-4o` model, optimizing it to generate rewrites that are more likely to result in preferred plans. The prompt used to prepare the data is as follows in Prompt:D.

Once again, we train the DPO-rewriter on `RECAP-train` using either the human or LLM based preference labels which corresponds to the `preferred_output` or `non_preferred_output`. The resulting model is used to generate rewrites with Prompt:B, and subsequently plans using Prompt:C.1 – as previously to maintain consistency – on the `RECAP-test` set.

We train the `gpt-4o-2024-08-06` model using OpenAI DPO fine-tuning, with hyperparameters `beta=0.1`, `n_epochs=3`, `batch_size=auto` and `learning_rate_multiplier=auto`.

### D.1    DPO:HUMAN DOWNSTREAM PERFORMANCE

After training the DPO model on the train data with human preference labels, we obtain the corresponding rewrite and plan (`DPO:human`) on `RECAP-test`. To restrict cost due to a cross product of comparison between rewriters, we only compare `DPO:human` plans with the best performing `Advanced` summarizer (from Table 2).

The results of this comparison using ranked analysis is shown in Figures 8a, 8b, 8c corresponding to intent-understanding challenge, topic, and conversation length respectively.

### D.2    DPO:LLM DOWNSTREAM PERFORMANCE

We repeat the same analysis, this time using `DPO:LLM` which is the rewriter model trained using LLM (`gpt-4.1` as it was the best performing model from Table 4) on `RECAP-test`. The results of the comparison between plans generated from `DPO:LLM` and `Advanced` rewriters is shown in Figures 9a, 9b, and 9c.

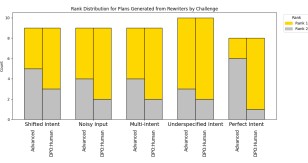

(a) Human Preference of Plans on `RECAP-test` by Intent Category

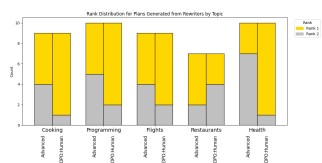

(b) Human Preference of Plans on `RECAP-test` by Topic

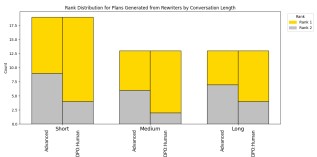

(c) Human Preference of Plans on `RECAP-test` by Conversation Length

Figure 8: Ranked Analysis: Human Preference of Plans generated between `Advanced` and `DPO:human` on `RECAP-test`: `DPO:human` ranks 1st across all intent categories, conversation lengths and most topics. Average pair-wise inter annotator accuracy: 64.3%.

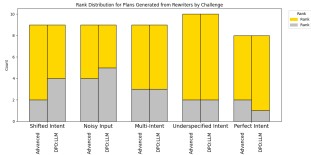

(a) Human Preference of Plans on `RECAP-test` by Intent Category

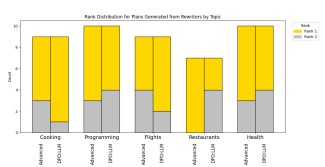

(b) Human Preference of Plans on `RECAP-test` by Topic

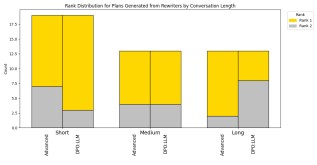

(c) Human Preference of Plans on `RECAP-test` by Conversation Length

Figure 9: Ranked Analysis: Human Preference of Plans generated between `Advanced` and `DPO:LLM` on `RECAP-test`: `DPO:LLM` ranks better for short conversation lengths and performs comparatively well across intent categories. Average pair-wise inter annotator accuracy: 61.48%.

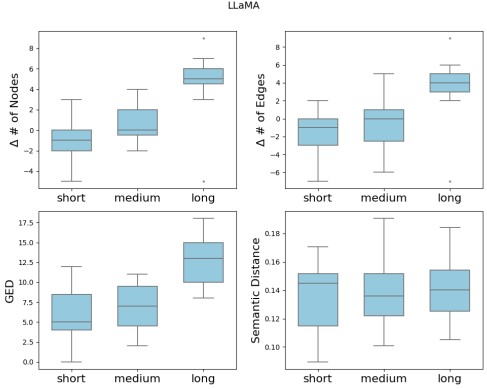

Figure 10: Sensitivity analysis for LLAMA

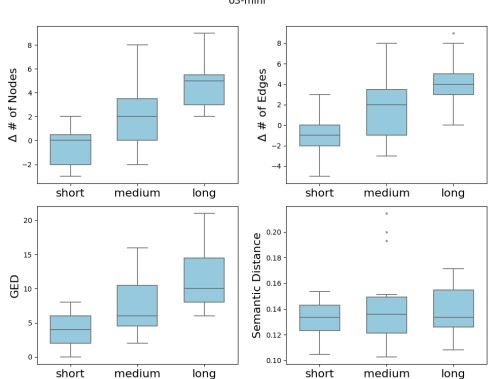

Figure 11: Sensitivity analysis for o3-mini

# E  SENSITIVITY ANALYSIS

## E.1  TOY DATSET CONSTRUCTION

To construct the toy dataset utilized for sensitivity analysis we generate USER-AGENT style conversations using `gpt-4o, temperature=0` using a prompt similar to A.2 without specifying explicit challenge instructions. The conversation length i.e. `conv_len` categories are defined as in section A.2.

## E.2  PLAN STRUCTURAL AND SEMANTIC SENSITIVITIES

Figure 12 shows that these plans diverge structurally, in terms of node/edge counts and graph edit distance, especially as conversation length increases, highlighting amplified variability in complex dialogues. While plan pairs typically show limited semantic variability, potentially due to the in-

ability of metrics like BERTScore to capture subtle distinctions, longer conversations tend to induce greater divergence.

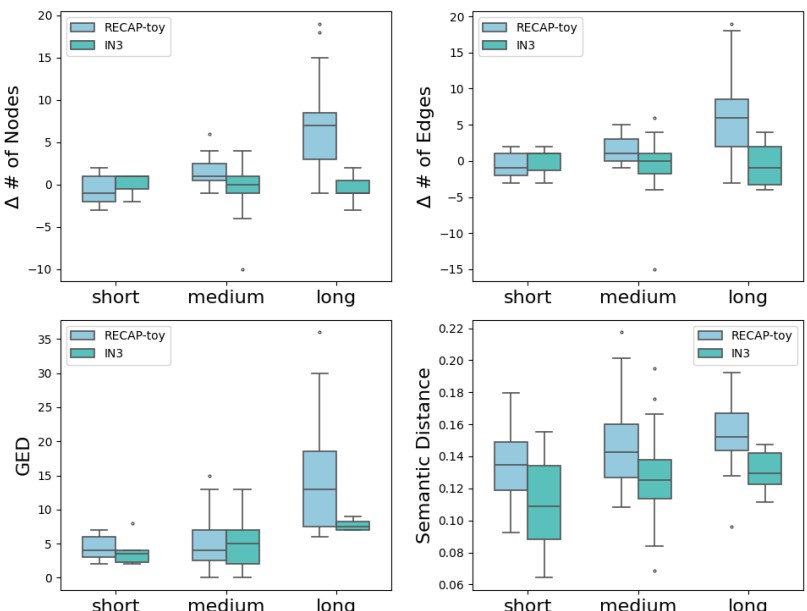

Figure 12: With similar setup to Table 1, the y-axis represents the semantic and structural differences between the resulting plans. The results indicate that the planner is indeed sensitive to variation in provided intents, with differences becoming more pronounced as conversations lengthen. RECAP-toy exhibits larger differences, suggesting that it is better suited for surfacing nuances in intent understanding than IN3

### E.3 SENSITIVITY ANALYSIS ACROSS PLANNERS

Although we use a *static* planner through our experiments, we extend our initial sensitivity analysis (Section 5.2) to various state-of-the-art LLM-based planners. This is done to perform a preliminary validation experiment that the results we see across our work is not a sole result of the planner quality we use i.e. GPT-4o.

We utilize the prompt defined in Section C.1 and employ LLaMA 3.3-70B with a temperature setting of 0 and GPT-o3-mini to generate plans using Dummy and Advanced rewriters on RECAP-toy data, as consistent with Section 5.2.

We use the same metrics defined in 3.2 to observe plan variation to input. Figures 10, 11 also show similar trends to GPT-4o (12) indicating that plan outputs are sensitive to the input characteristics – output of the rewriter.

## F  RECAP BENCHMARK

We release 810 conversations as the RECAP benchmark. In our experiments, due to cost and effort constraints because of human annotation, we only utilized 150 of these conversation instances, maintaining uniformity across conversation lengths, topics and intent-challenge categories.

**Stats**  The RECAP dataset is uniformly distributed across five distinct topics — *cooking*, *programming*, *flights*, *restaurants*, and *health* — with 162 instances each. Similarly, the intent_category dimension covers the different intent-understanding related categories: *shifted_intent*, *noisy_input*, *underspecified_intent*, *multi_intent*, and *perfect_intent*, also with 162 in-

stances each. Conversation lengths (`conv_len`) are evenly distributed across three buckets: *short* (270), *medium* (270), and *long* (270), ensuring balance across all dimensions.

**Vetting**   The synthetically generated conversations are vetted for adherence to instructions, overall coherency, and to ensure no bias or malicious content is present. Personal information such as names, contact details, including phone numbers and email addresses (even if generated by the LLMs, serving as placeholders), were redacted.

Table 7: Win/Tie/Loss percentage for plans generated from `DPO:human` vs `Advanced` across intent categories

| Intent Category | Win Rate | Tie Rate | Loss Rate |
|---|---|---|---|
| Shifted Intent | 35.33 | 40.00 | 24.67 |
| Noisy Input | 26.67 | 47.33 | 26.00 |
| Multi-Intent | 24.00 | 46.00 | 30.00 |
| Underspecified Intent | 22.00 | 54.00 | 24.00 |
| Perfect Intent | 26.00 | 35.33 | 38.67 |
| **Total** | 26.80 | 44.53 | 28.67 |

**Evaluation**   Using the best-performing fine-tuned evaluator (Table 4), we evaluate the plans generated on the entire RECAP dataset. The plans are generated using `DPO:human` and `Advanced` rewriters, utilizing the planner described in Section C.1. The results are shown in Table 7, where Win Rate denotes the plan from `DPO:human` was preferred to `Advanced` rewriter, and Loss Rate denotes vice versa. We observe there is largely neutral preference across intent categories.

