# OpenReview forum: "RECAP: REwriting Conversations for Intent Understanding in Agentic Planning"
_ICLR.cc/2026/Conference — ICLR 2026 Conference Withdrawn Submission_

### Official Review · Reviewer_SzQy · 2025-10-23

**Soundness:** 1
**Presentation:** 2
**Contribution:** 1
**Rating:** 2
**Confidence:** 4

**Summary:**

This paper introduces **RECAP**, a benchmark designed to evaluate *intent rewriting* in open-ended, multi-turn dialogue systems, especially those involving agentic planning. The authors argue that existing dialogue benchmarks fail to capture the process of reformulating ambiguous or underspecified user intents, which is crucial for accurate downstream task planning. The work includes both a dataset (RECAP) and a lightweight environment (RECAP-Toy) for controlled experiments. The study further explores whether fine-tuning a large language model (GPT-4-class) on preference pairs can improve intent rewriting and, consequently, planning performance.

While the motivation is relevant, namely linking intent understanding with planning, the methodology, evaluation, and contribution boundaries remain unclear.

**Strengths:**

- **Problem framing**: The paper identifies 'intent rewriting' as a potential bottleneck in multi-turn, open-ended dialogue systems, especially where goal-oriented planning is required.
- **Benchmark focus:** Establishing a benchmark around this specific functionality (rewriting for planning) is an interesting step toward formalizing a neglected capability of LLM-based agents.
- **Initial empirical results:** The study provides preliminary evidence that better intent rewriting can positively influence plan generation quality.

**Weaknesses:**

1. **Unclear novelty and positioning**
   The contribution of RECAP as a benchmark is insufficiently distinguished from existing datasets. Intent rewriting overlaps conceptually with prior dialogue understanding and task reformulation benchmarks (e.g., MultiWOZ, ALFRED, Tau-Bench). The authors should clarify what new capability RECAP measures that others cannot.

2. **Synthetic and limited data generation**
   The benchmark heavily relies on **synthetic dialogues**, raising concerns about representativeness and realism. Comparable or richer examples could be derived from existing corpora, reducing the need for a fully synthetic setup.

3. **Questionable need for RECAP-Toy**
   The introduction of a simplified environment (RECAP-Toy) seems to distract from the main benchmark. Its utility is unclear, and it reduces the ecological validity of the evaluation.

4. **Incomplete evaluation design**
   - The evaluation lacks an end-to-end metric (e.g., task success or goal completion) connecting rewriting quality to actual execution outcomes.
   - Reported metrics (pairwise win rates, plan quality improvement) remain qualitative and fail to quantify the true downstream impact.
   - The paper assumes a rigid sequential flow, from *intent understanding* to *planning* and then *execution*, without exploring reactive or interleaved strategies that may render rewriting unnecessary.

5. **Narrow generalization and static planner assumption**
   All experiments are conducted within a single planning setup. It remains unclear whether improvements generalize across planners, domains, or conversation styles.

6. **Limited interpretability of findings**
   The main takeaway, showing that intent rewriting affects planning, is intuitively expected but not operationalized into actionable insights. The proposed methods (prompt engineering, preference fine-tuning) are well-known and not evaluated for robustness or transfer.

**Questions:**

- **Overall design:** Can user interaction occur in parallel with plan execution, or must all ambiguities be resolved beforehand? This affects the necessity of intent rewriting.
- **Figure 3:** Is there a rationale behind the specific topic arrangement? Clarifying the visual structure would help interpretability.
- **Line 199:** Please clarify what is meant by “pair-wise method” in the context of preference training.
- **Footnote 3:** The reviewer still believe that including a basic rewriter baseline would provide useful insights and facilitate a more comprehensive evaluation of the results.
- **Line 326:** The purpose of RECAP-Toy requires clearer justification, namely what gap does it fill beyond debugging convenience?
- **Lines 331–333:** The sentence appears duplicated; editing needed.
- **Lines 364–372:** The authors acknowledge a flaw in the benchmark but do not fix it before release. This undermines the dataset’s reliability.
- **Table 5:** For DPO:LLM on *Shifted Intent*, *Noisy Input*, and *Total*, the **bold font** should highlight the **Loss Rate**.

---

> ### Author Response · Authors · 2025-11-20
> **Addressing Questions**
>
> We thank the reviewer for their questions and answer them as follows:
> 1. **Overall Design** In our current set-up, focus is on intent understanding and its downstream application in agentic planning. For this reason, we aim to resolve ambiguities beforehand. Interleaved ambiguities may as well be a different set-up and approach, possibly useful in some but not in all cases eg. where end user is not directly working with a plan but only with a dialog module – the dialog module should capture intent and pass down to planner.
> 2. **Figure 3**: We include topics utilized for general purpose task-oriented conversations, also inspired from existing works like MultiWOZ. We generate a uniform distribution across topics (apart from challenges and conversation lengths).
> 3. **L199**: To compare different rewrites and their planning outcomes, we evaluate the structural, semantic & plan-preference metrics between two plans. Details of this pair-wise evaluation methodology is described in Appendix C.2
> In the context of preference training, we utilize DPO training (Section 4.2) whereby the preferred plan is treated as the _positive_ sample and the non-preferred plan is the _negative_ sample (L275-277).
> 4. **Footnote 3**: We point out to the reviewer that the performance of the basic rewriter is computed for our main results (Table 2, Table 3) showing its performance relative to the Dummy and Advanced rewriters. For the sensitivity analysis set-up we deliberately omit the Basic rewriter as the aim of the analysis is to only show contrasting planning outcomes on differing input, for purposes of initial experiments. The detailed pair-wise comparison of different rewriters is described in Section 5.3
> 5. **L326**: As we mentioned while addressing the weaknesses, RECAP-toy is only utilized for a small-scale sensitivity analysis to compare and contrast the sensitivity of multiple planners to our dataset vs. the IN3 dataset (Section 5.2, L325-327). By no means is it utilized for the remainder of the experiments and results.
> 6. **Line 331-333**: We thank the reviewer for pointing out the typo, which we shall address.
> 7. **L364-367**: We would like to point out to the reviewer that the “fake intent shifts” were **deliberately** included by us to enrich the benchmark (L365-367), and hence is not a flaw. The purposes for such instances is further elaborated in Appendix B.1, and through the results we note that even an advanced rewriter may sometimes fail to capture these nuanced challenges of fake intent shifts.
> 8. **Table 5**: To clarify, In Table 5, we bold the cases where our DPO rewriter (DPO:LLM or DPO:human) outperforms the advanced rewriter. We observe that DPO:human consistently performs better than the Advanced rewriter.  We note the reasons why DPO:llm may not be doing so in all cases (L 431-434).

---

> ### Author Response · Authors · 2025-11-20
> **Addressing weaknesses**
>
> We appreciate the reviewer’s feedback and questions. We address the weaknesses raised by the reviewer below:
>
> 1. The primary focus of RECAP is on capturing more challenging, complex (user-assistant) conversations, which introduces the notion of intent rewriting as a crucial step. Previous works eg. MultiWOZ have a narrower focus on slot-filling and completion (Section 1, L079-082), with limited insights into how such complex conversations affect downstream agentic planning, a highly relevant topic in the community.
>
> 	We further establish the clear motivation behind constructing RECAP in Section 3 (ref: L156-161), by introducing **varying challenges**, such as multi-intent, vague intent, intent shift, within the dataset and across **domains, conversation lengths to enhance generalization**.
>
> 	In Section 5.2, we demonstrate the **effectiveness of RECAP** compared to the existing IN3 dataset, which itself is derived from MultiWOZ. L336-L339 summarize our findings where IN3 fails to capture complex intent-understand challenges, which RECAP improves upon.
>
> 2. Following the previous point, we establish how existing benchmarks fail to capture more nuanced challenges within intent-understanding. As such, we aimed to curate complex conversations of varying difficulty, topic, and length. In Section 3.1 (L183-185) we discuss the **benefits, in terms of cost and scalability, to utilize synthetically generated conversations**, further grounding our methodology on existing works which demonstrate the promise of LLM-generated synthetic data especially for intent understanding tasks (Efficacy of Synthetic Data as a Benchmark,  Maheshwari et al., 2024; Evaluating Language Models as Synthetic Data Generators, Kim et al., 2024).
>
> 	Furthermore, the RECAP dataset is then **human vetted** (Section 3.1, L189-192, Appendix F L1245-1248) for coherency and adherence to prompt instructions, which aims to follow the characteristics of natural human conversations. Moreover, we also include an intent category **“Noisy input”** specifically to mimic real-world dialogues (Section 2, L129-134).
>
> 	Qualitative instances of conversations are included in Table 6, demonstrating the richness of RECAP benchmark in terms of diversity, complexity for the required task.
>
> 3. We would like to clarify that **RECAP-toy is not an environment**, but a small subset, **solely used for initial experiments on planner sensitivity (Section 5.2)** and is not the main focus of our work. In Section 5.1 L299-305 we clearly describe the subset of data utilized for the main results of our paper (Tables 2, 3, 4, 5) which is independent of RECAP-toy. We hope these clarifications help address any possible misunderstanding on this point
>
> 4. As pointed out in Section 3.2 (L213-215) we utilize a preference metric as an ultimate measure of utility to end-users. In our setting, we do not focus on plan execution or interleaved strategies, which may be directions of future work.
> We would like to further point out that our setting is beneficial for several cases eg. interaction of the user with a chat system for restaurant recommendations, where the planning/execution steps are blackboxed to the user, as such interleaved planning-clarification steps are not feasible. In such settings where users only state their intent (textually), we believe intent rewriting forms as essential step before downstream planning.
>
>
> 5. A static planner is **key** to our experiment set-up. In Section 5.1 (L290-293) we introduce the reasoning behind utilizing a static planner to solely evaluate the effect of intent rewriting. If our planner were to be dynamic in nature, we would be unable to trace the differences in the planning outcomes back to the intent-rewriting step alone. Our experimental setup assumes that utilizing the same model, prompt, hyperparameters we limit the variability of the planner module, and as such can reliably evaluate the differing plans due to the input (intent rewrite).
> Nevertheless, we also **conduct our sensitivity analysis (Section 5.2) on different planning modules** (L339-342) utilizing Llama, o3-mini models which are illustrated further in Figures 10, 11.
>
> 6. We thank the reviewer for feedback on utilizing our models and fine-tuning strategies for transfer, which we believe would be interesting future work. However, this work demonstrates that **feeding raw conversations to a planner produces sub-par plans** (eg. Figure 12, whereby plans from different rewrites result in highly different planning outcomes). While this may seem intuitive, to our knowledge **RECAP is the first work which quantitatively aims to evaluate this** and offers insightful solutions to using more advanced, fine-tuned rewrites for such a task. We believe the research community will benefit from a benchmark which is aimed to enhance planning capabilities with simple, yet effective rewriting strategies (Table 2, Table 5).

---

> > ### Comment · Reviewer_SzQy · 2025-11-27
> > **Reviewer response to the authors’ comments**
> >
> > I thank the authors for their detailed responses. I must note, however, that I found it somewhat difficult to connect several of the replies directly to the concerns I raised.
> >
> > In summary, while I appreciate the authors’ perspective, I remain concerned that relying on a static planner and omitting end-to-end evaluation metrics substantially limits the contribution. Focusing primarily on intent reformulation places this work close to a large existing body of research on synthetic data generation for intent detection and query expansion.
> >
> > Nonetheless, in light of the authors’ clarifications, I will raise my score.

---

> > > ### Author Response · Authors · 2025-11-27
> > >
> > > We greatly appreciate the reviewer for their thoughtful response and comments. We believe there are some key clarifications left to address, which we do so as follows:
> > >
> > > 1. > ...intent reformulation places this work close to a large existing body of research on synthetic data generation for intent detection and query expansion
> > >
> > > Existing works, such as MultiWOZ focus on slot-filling for intent understanding, which makes the focus of their work on understanding intent in terms of certain classification slots (eg. **select** the arrival time, **inform** about (restaurant) open-hours, **recommend** restaurant). Even ALFRED, introduces simple but clearly defined user goals eg. "Pick up the toilet paper roll and put it by the toilet.". However, none of these works focus on capturing nuanced challenges faced within intent understanding, which we address: for instance, if the user intent within a conversation shifts from searching a hotel to booking a flight (_intent shift_) or if the user provides vague instructions or requests for hotel search (_underspecified intent_).
> > >
> > > To our knowledge, **RECAP** is the first benchmark which introduces multiple challenges faced in intent understanding (eg. _multi intent_, _underspecified intent_, _intent shift_) across domains with varying lengths in conversation, to introduce generalizability. We compare the performance of IN3 dataset (itself derived from MultiWoz) versus our RECAP benchmark in L336-L339 summarizing our findings (Section 5.2, Table 1). Our analysis shows that RECAP is more effective than existing datasets, as it introduces realistic and complex scenarios which surface the need for more sophisticated intent reformulation.
> > >
> > > 2. > relying on a static planner
> > >
> > > We thank the author for their comment but underline that this is a key concept to understand the set-up of our work. Our experiments in Section 5.3, 5.4, 5.5. introduce a pair of rewrites (i.e. different intent formulations of the same USER-AGENT conversation), with **our objective being to show that different intent formulations have drastic differences in outcome of the plans**. If we were to introduce a dynamic planner with the inclusion of reasoning or interleaved steps, this prevents us from determining the **sole cause** of the planning outcome (whether it was due to the rewrite or reasoning capabilities / prompt differences of the planner). For this very reason, we treat the planner as a static one, whereby the outcomes in planning can be traced back to the intent rewrite (the input to the planner).
> > >
> > > Furthermore, due to intent reformulation itself, we note structural, semantic differences in the planning outcomes (Table 3) and preference based differences in Table 2. As such, we note a **static** planner to be essential to our experiment set-up in this work, and results using a dynamic planner would be unsuitable and unsound for our scenario, and leave this for future work to explore.
> > > _Note_: for our sensitivity analysis (Section 5.2) on different planning modules (L339-342) utilizing Llama, o3-mini models which are illustrated further in Figures 10, 11
> > >
> > > 3. >omitting end-to-end evaluation metrics
> > >
> > > In this work, we utilize a **preference metric** as the key downstream evaluation of a plan (Section 3.2, L214-L215). While this may not be the most ideal form to measure task/goal success, it actually allows us to capture if the user intent was actually met by the planner or not.
> > >
> > > Let's consider a scenario where a user interacts with a restaurant booking system to reserve a non-smoking table for 4 people at an upscale restaurant. Through the conversation, the user also states they would prefer Italian cuisine which also sells coffee. Consider two intent formulations (one meeting the user requirements, while the other does not): in such a scenario even task/goal execution success (which if we assume to be "book restaurant table") is met under both scenarios. In order to capture the actual user intent, the success metric needs to be more complex eg. defining a sophisticated environment whereby plan execution allows us to confirm if the restaurant booked meets the requirements stated by the user. In this paper, we made a **simple assumption of utilizing downstream user preference** (utilizing expert annotators and LLMs) as the end-to-end metric, in lieu of a more complex execution driven metric, which was beyond the scope of our work.
> > >
> > >
> > > We sincerely hope these further clarifications are able to address some key concerns raised by the reviewer. We re-iterate the effectiveness of RECAP for more complex intent-understanding challenges, the use of expert/LLM judges for our downstream evaluation and the nature of our research set-up which makes a static planner key to our design and experiments.

---

### Official Review · Reviewer_8jrh · 2025-10-30

**Soundness:** 2
**Presentation:** 3
**Contribution:** 2
**Rating:** 2
**Confidence:** 4

**Summary:**

The paper focuses on intent rewriting to help planners understand ambiguous, long conversations. They propose a new benchmark, RECAP, to test this. They show that a prompt-based rewriter (Advanced) leads to better plans than naive baselines. Further, they DPO-fine-tune rewriter models using both human labels and labels from an automatic LLM-evaluator.

**Strengths:**

1. The work clearly identifies that user intent can change during a conversation which can confuse agentic planner. I like the idea of decomposing the task, clearly this is better than simply generating summary of the conversation.
2. The authors introduce a new dataset specifically designed to capture these conversational challenges (shifted intent, multi-intent, etc.). Their sensitivity analysis also shows it's more challenging and is better at distinguishing between a dummy rewriter and advanced rewriter.

**Weaknesses:**

Main:
1. The work assumes that a better plan will lead to successfully completing a task. They do not evaluate this hypothesis and the work is missing downstream evaluation.
2. It is a synthetically generated dataset, which might not work for human conversations. I think synthetic data is useful if you can show that it helps in tasks that involve real-human conversations.
3. The human evaluation is done on a small subset, and shows different results than LLM-as-a-judge results on the complete set -- showing that DPO:human has no significant gain over the advanced rewriter.
4. The work uses a single static planner. It remains to see how this module would help other models. Would this rewriter be useful for new models like GPT-5. An experiment to show this could be trying different parameter models from the same family or comparing cross family of models.

**Questions:**

1. What is the llm-as-a-judge performance on RECAP-toy?
2. What is the sensitivity analysis for basic vs advanced?
3. Why are the llm-as-a-judge evaluation so different than human judgements? What are human judges catching that the llm-as-a-judge doesn't?

---

> ### Author Response · Authors · 2025-11-20
>
> We appreciate the reviewer’s feedback and questions. We address the weaknesses raised by the reviewer below:
>
> 1. In this work, we assume a setting of a user-assistant conversation whereby the downstream utility is measured through plan preference. Although plan execution will surely be an interesting line of future work, in this paper our findings demonstrate how capturing better intent leads to better plans for the end-user. It is intuitive imagine cases (eg. user stating requests for improving health) whereby a generated plan serves as an output to the end-user, and we measure if the user prefers the generated plan or not, based on their intent and request.
>
> 2. In Section 3.1 (L183-185) we discuss the benefits, in terms of **cost and scalability**, to utilize synthetically generated conversations, further grounding our methodology on existing works which demonstrate the promise of LLM-generated synthetic data especially for intent understanding tasks (Efficacy of Synthetic Data as a Benchmark,  Maheshwari et al., 2024; Evaluating Language Models as Synthetic Data Generators, Kim et al., 2024).
>
> 	Furthermore, the RECAP dataset is then **human vetted** (Section 3.1, L189-192, Appendix F L1245-1248) for coherency and adherence to prompt instructions, which aims to follow the characteristics of natural human conversations. Moreover, we also include an intent category **“Noisy input”** specifically to mimic real-world dialogues (Section 2, L129-134).
>
> In Section 5.2, we further demonstrate the effectiveness of RECAP compared to the existing IN3 dataset, which itself is derived from MultiWOZ. L336-L339 summarize our findings where IN3 fails to capture complex intent-understand challenges, which RECAP improves upon.
>
> 3. We respectfully point out to the reviewer that our human evaluation was limited to a smaller set due to **cost constraints** (L299-300). Through our results using expert annotators (Section 5.5, Table 5) we demonstrate **significant improvements of using the DPO model** compared to Advanced. The results summarized in Table 7 utilizes LLM-as-a-judge which is shown (from Table 4) to have only ~65% accuracy compared to human annotations. As such, it is expected that the LLM-as-a-judge fails to correctly predict several preference instances. We respectfully argue that this point does not serve as a weakness of our work, as we explore the benefits of DPO:human in more suited conditioned experiments.
>
> 4. A static planner is **key** to our experiment set-up. In Section 5.1 (L290-293) we introduce the reasoning behind utilizing a static planner to solely evaluate the effect of intent rewriting. If our planner were to be dynamic in nature, we would be unable to trace the differences in the planning outcomes back to the intent-rewriting step alone. Our experimental setup assumes that utilizing the same model, prompt, hyperparamaters we limit the variability of the planner module, and as such can reliably evaluate the differing plans due to the input (intent rewrite).
> Nevertheless, we also **conduct our sensitivity analysis (Section 5.2) on different planning modules** (L339-342) utilizing Llama, o3-mini models which are illustrated further in Figures 10, 11. We'd be happy to include performance with planners utilizing newer models in a followup version.

---

> ### Author Response · Authors · 2025-11-20
>
> We thank the reviewer for their questions and answer them as follows:
>
> 1. RECAP-toy, is a small subset of RECAP, and  is only used for sensitivity analysis (Section 5.2) and uses human evaluators only. The performance of LLM-asa-judge on the RECAP subset is summarized in Table 4.
>
> 2. We point out to the reviewer that the performance of the basic rewriter is computed for our main results (Table 2, Table 3) showing its performance relative to the Dummy and Advanced rewriters. For the sensitivity analysis set-up we deliberately omit the Basic rewriter as the aim of the analysis is to only show contrasting planning outcomes on differing input, for purposes of initial experiments.
>
> 3. As shown in Table 4, the base LLMs-as-a-judge perform poorly compared to humans on the plan preference task. This is due to LLMs known to being inefficient and biased evaluators (Large Language Models are not Fair Evaluators, Wang et al., 2024; Large Language Models are Inconsistent and Biased Evaluators, Stureborg et al., 2024). We show that fine-tuning the base LLMs improves performance (Table 4) significantly. It can be hypothesized that LLMs as a judge often fail to capture the nuanced complexities of user-intent and evaluating a downstream Plan DAG compared to expert annotators.

---

### Official Review · Reviewer_YqBY · 2025-10-31

**Soundness:** 2
**Presentation:** 3
**Contribution:** 2
**Rating:** 4
**Confidence:** 3

**Summary:**

This paper introduces RECAP, a benchmark designed to evaluate and improve intent rewriting for large language model (LLM)-based agentic planning. The authors argue that existing intent classification methods struggle with ambiguity, drift, and multi-intent scenarios in real-world dialogues. RECAP contains 810 synthetic user–agent conversations covering diverse topics and intent challenges. The paper also proposes an LLM-based evaluator for assessing plan quality derived from rewritten intents. Through experiments comparing prompt-based and DPO-finetuned rewriters, the study shows that effective intent rewriting can improve downstream task planning quality and alignment with human preferences.

**Strengths:**

- This paper is easy to read and logically clear.
- The experimental framework is relatively systematic, with complete links.
- This paper is clearly presented with descriptive figures.

**Weaknesses:**

- The experimental scope is limited: although RECAP contains 810 dialogues, only 150 are used for evaluation, without disclosure of their composition across intent types, domains, or dialogue lengths. This weakens the representativeness of the results and prevents meaningful analysis of how dialogue length or topic affects model performance.
- The paper lacks a quantitative evaluation of intent understanding. Scenario categories such as Shifted, Noisy, and Underspecified Intent are described only by examples, not by formal definitions or measurable criteria. Without explicit intent-level metrics or annotation guidelines, it is unclear whether the reported improvements truly reflect better intent recognition.
- The evaluation framework conflates planning quality with overall task success. Results are based mainly on plan preferences judged by humans or LLMs using a single planner, without statistical significance tests or execution-based validation. This limits the reliability and generalizability of the claimed improvements.

**Questions:**

- How were the 150 dialogues selected from the 810 available in RECAP? Could you provide the distribution of these samples across intent types, dialogue lengths, and topics?
- How do you define and annotate the five intent challenge types (Shifted, Noisy, Underspecified, Multi-Intent, Perfect)? Are there formal annotation criteria or inter-annotator agreement statistics?
- Since intent understanding is the main focus, do you have any quantitative metric—such as intent classification accuracy or intent drift detection rate to complement plan preference evaluation?
- Have you tested whether the observed improvements hold across different planners or decoding settings (e.g., beyond GPT-4o)?
- Can you provide statistical significance tests or confidence intervals for human evaluation results to demonstrate the robustness of the reported improvements?

---

> ### Author Response · Authors · 2025-11-20
>
> We appreciate the reviewer’s feedback and questions. We address the weaknesses raised by the reviewer below:
>
> 1. As mentioned in Section 5.1 L299-300, we **uniformly sample the 150 dialogues from the entire RECAP dataset**, maintaining distribution across the intent challenges (30 each), lengths (50 each) and topics (30 each) (ref: Appendix F). As such, the results on the subset reflect the distribution on the entire dataset.
> Furthermore, we provide detailed analysis of how **category, length, topic affect the model performance in Figure 7 (Dummy, Basic, Advanced rewriters), Figure 8 (DPO:human), and Figure 9 (DPO:llm)**, with the respective captions summarizing the key take-aways.
> The details on the analysis set-up is provided in Appendix C2.2 (L1045 onwards).
> 2. In this work, we provide explicit instructions to the LLM (as detailed in Appendix A.2) to generate conversations adhering to the prompt, based on challenge, topic and length. Furthermore these conversations are **human vetted** (Section 3.1, L189-193) to ensure they remain true to the instructions.
> The definitions for these instructions is included in our supplementary material (src/prompts.py) as follows:
> ```
>     "intent_shift": "The user's intent should shift partway through the conversation."
>     "noisy_data": "Include real-life noise such as greetings, small talk, or irrelevant details."
>     "underspecified_intent": "The user's goal should be vague or incomplete, requiring inference."
>     "multi_intent": "The user should express multiple distinct intents, possibly in the same turn."
>     "perfect_intent": "The user's goal should be clear, unambiguous, and fully specified."
> ```
>
> Although previous works focus on intent understanding in terms of slot-filling (Section 1, L079-082), we aim to capture more nuanced and complex challenges in intent understanding. In our study setting, we achieve this by generating conversations reflecting such complexities (intent challenge) and variations (in terms of length, topics).
> Evaluating such complex intent understanding may definitely be a focus of future work, however **our work acts as a stepping stone in terms of holistic intent-understanding across varied challenges**.
>
> 3. Section 5.1 (L290-293) motivates the reasoning behind utilizing a **static planner, in our study, as a key component to solely evaluate the effect of different rewriting techniques on planning outcomes**.
> We believe our work is beneficial in cases where an end-user would look to review the plan after stating their requests and clarifications to an assistant, which is measured through plan preference as a measure of downstream utility and user satisfaction. Plan execution for tasks is a more objective study, and can be explored in future works.
>
> Details regarding the **inter-human agreement scores** is provided in the captions of Figures 7, 8, 9. We agree that such preferences are subjective and may involve several nuances. However, we believe that in our setting, semantic, structural and preferential metrics (Section 3.2) capture the different planning outcomes due to rewrites and the initial user satisfaction level.

---

> > ### Author Response · Authors · 2025-11-20
> >
> > We thank the reviewer for their questions and hope our previous comment addressed several of them. Furthermore,
> >
> > - We would like to clarify that our work moves away from traditional intent understanding as a classification/slot-filling task (as motivated in Figure 1). As such we employ structural, semantic, preferential metrics (Section 3.2) to capture the variations in planning outcomes. Preferential metric serves as the measure to evaluate the nuanced and challenging intent-understanding task in this scenario. We appreciate the suggestion of evaluations being more fine-grained, which can be explored in future work.
> >
> > - For our study, a static planner was key to our experimental set-up. Nevertheless, we also explore the sensitivity analysis of planners using Llama and o3-mini models, as described in L339-342; illustrated further in Figures 10, 11.
> > Human evaluation pairwise inter annotator accuracy is reported in Figures 7, 8, 9 (kindly refer to the captions). We acknowledge the relative subjectivity and difficulty of such a nuanced task.

---

### Official Review · Reviewer_eAtd · 2025-11-01

**Soundness:** 3
**Presentation:** 3
**Contribution:** 3
**Rating:** 6
**Confidence:** 4

**Summary:**

This paper introduces the RECAP benchmark for evaluating intent rewriting in LLM-driven dialogue systems. This benchmark captures key challenges such as ambiguity, intent drift, and mixed-goal conversations. By reformulating the dialogue into a concise representation of the user's goal, rewriting enables more accurate downstream task execution planning.Experimental results demonstrate that both prompt-based and DPO-trained rewriters significantly improve planning effectiveness. These results provide guidance for building more effective and adaptive dialogue agents.

**Strengths:**

1. RECAP is a new benchmark dataset specifically designed to evaluate and improve intent rewriting, especially in open-domain dialogue systems.
2. The RECAP systematically captures multiple intent rewriting challenges, including intent ambiguity, intent drift, ambiguity, and multi-objective dialogue.

**Weaknesses:**

1.The RECAP dataset spans five distinct domains: cooking, programming, flights, restaurants, and health. These topics were selected to cover a diverse range of common, goal-oriented conversational scenarios that users frequently engage in with AI assistants.While these domains are representative of many real-world interactions, the areas covered are still not comprehensive enough to prove more generalization.

2.RECAP categorizes intent challenges into five types: shifted intent, noisy input, underspecified intent, multi-intent, and perfect intent. While these categories cover a broad range of intent-related challenges, other types such as contradictory intent (where users express conflicting goals) or emotionally charged intent (where emotional language affects goal clarity) may also be relevant.

3.The paper's core experiments used only 150 data points, representing only about 18.5% of the entire RECAP dataset (810 instances). This raises questions about the statistical significance and reliability of the experimental results. Is a method that performs well on a small subset also robust on the full dataset? The paper's appendix mentions an evaluation of DPO:human and Advanced on the full dataset, but the results show a very neutral preference rate, which is somewhat inconsistent with the paper's conclusion that DPO:human is significantly superior to Advanced. This requires further explanation.

**Questions:**

1.The RECAP dataset is constructed across five specific domains: cooking, programming, flights, restaurants, and health. Could the authors further elaborate on the criteria and rationale for selecting these particular themes? To what extent can these themes represent common task-planning scenarios in open-domain dialogue assistants? How well can the conclusions drawn from performance on these themes generalize to other important domains (e.g., finance, education, entertainment)? Have the authors considered the potential biases introduced by this domain selection?

2.The paper defines five categories of intent challenges: shifted intent, noisy input, underspecified intent, multi-intent, and perfect intent. Why were these five types chosen as the core challenges for the benchmark? Are there other equally important and common intent-understanding challenges (e.g., contradictory intents, implicit intents, or intents obscured by emotional language) that are not covered by the current taxonomy? Could the authors provide a more detailed justification that the current five categories are both necessary and sufficient for evaluating planning utility?

---

> ### Author Response · Authors · 2025-11-20
>
> We thank the reviewer’s feedback and questions. We address them below:
>
> 1. While we acknowledge the limitations of our benchmark, we respectfully argue that this point does not reflect a weakness in our work. We believe **RECAP provides a diverse and generalized view on several intent-understanding challenges, topics and conversation lengths** which was not explicitly explored in existing literature. Regarding the topics used in our benchmark, we take inspiration from existing works (such as MultiWoz, IN3) which focus on similar domains. For comparison, even MultiWoz itself covers only 7 domains namely restaurant, hotel, attraction, taxi, train, hospital, police.
>
> We further perform an analysis on model performance based on the topics (in Figures 7, 8, 9). We hope these analyses and careful benchmarking with a view for generalizability is considered by the reviewer.
>
> 2. We do not claim that the current benchmark’s intent categories are complete, however are effective. We assert that RECAP, to our knowledge, is the first benchmark which explicitly aims to capture nuanced aspects of intent-understanding and groups them according to some common issues. This is further **demonstrated through existing benchmarks’ inability to capture such nuanced challenges through their data** (L336-L339).
> We also perform analysis on how **model performance varies with intent challenges** (Table 3, Section 5.3 L353-372). We believe this serves as a platform for future works to work and expand upon, exploring more categories and domains, and respectfully suggest that such points do not reflect a weakness in our method or results.
>
> 3. The core experiments requiring human evaluation was limited to a smaller set due to **cost constraints** (L299-300). In Section 5.1 L299-300, we describe that we **uniformly sample** the 150 dialogues from the entire RECAP dataset, maintaining distribution across the intent challenges (30 each), lengths (50 each) and topics (30 each). As such results from this subset reflect the distribution across the entire data.
>
> Through our results using expert annotators (Section 5.5, Table 5) we demonstrate **significant improvements of using the DPO model** compared to Advanced. The results summarized in Table 7 utilizes LLM-as-a-judge which is shown (from Table 4) to have only ~65% accuracy compared to human annotations. As such, it is expected that the LLM-as-a-judge fails to correctly predict several preference instances. This can be hypothesized due to LLMs known to being inefficient and biased evaluators (Large Language Models are not Fair Evaluators, Wang et al., 2024; Large Language Models are Inconsistent and Biased Evaluators, Stureborg et al., 2024).

---

### Comment · Area_Chair_mLX1 · 2025-11-28
**Please Check the Authors' Responses**

Dear Reviewers,

The authors have posted their responses. Could you please take a moment to review their responses and check whether your concerns have been adequately addressed (if you have not done it yet)? If possible, kindly initiate the discussion at your earliest convenience.

Your timely assistance is essential for keeping the review process on track. Thank you very much for your support and contribution.

Best regards, Your AC

---

### Note · Authors · 2025-12-12

I have read and agree with the venue's withdrawal policy on behalf of myself and my co-authors.